# Molecular basis of maintaining an oxidizing environment under anaerobiosis by soluble fumarate reductase

Sunghwan Kim[1,7], Chang Min Kim[2], Young-Jin Son[1], Jae Young Choi[3], Rahel K. Siegenthaler[4], Younho Lee[5], Tae-Ho Jang[1], Jaeyoung Song[1], Hara Kang[6], Chris A. Kaiser[4] & Hyun Ho Park[2]

Osm1 and Frd1 are soluble fumarate reductases from yeast that are critical for allowing survival under anaerobic conditions. Although they maintain redox balance during anaerobiosis, the underlying mechanism is not understood. Here, we report the crystal structure of a eukaryotic soluble fumarate reductase, which is unique among soluble fumarate reductases as it lacks a heme domain. Structural and enzymatic analyses indicate that Osm1 has a specific binding pocket for flavin molecules, including FAD, FMN, and riboflavin, catalyzing their oxidation while reducing fumarate to succinate. Moreover, ER-resident Osm1 can transfer electrons from the Ero1 FAD cofactor to fumarate either by free FAD or by a direct interaction, allowing de novo disulfide bond formation in the absence of oxygen. We conclude that soluble eukaryotic fumarate reductases can maintain an oxidizing environment under anaerobic conditions, either by oxidizing cellular flavin cofactors or by a direct interaction with flavoenzymes such as Ero1.

[1] New Drug Development Center, Daegu-Gyeongbuk Medical Innovation Foundation, Daegu 41061, South Korea. [2] College of Pharmacy, Chung-Ang University, 06974 Seoul, South Korea. [3] Department of Biochemistry, Yeungnam University, Gyeongsan 38541, South Korea. [4] Department of Biology, Massachusetts Institute of Technology, Cambridge, MA 02139, USA. [5] College of Pharmacy and Yonsei, Institute of Pharmaceutical Sciences, Yonsei University, 85 Songdogwahak-ro, Yeonsu-gu, Incheon 21983, Republic of Korea. [6] Division of Life Sciences, College of Life Science and Bioengineering, Incheon National University, Incheon 22012, Republic of Korea. [7] Present address: R&D Center, Polus Inc., 9 Songdomirae-ro, Yeonsu-gu, Incheon 21984, South Korea. These authors contributed equally: Sunghwan Kim, Chang Min Kim, Young-Jin Son. Correspondence and requests for materials should be addressed to S.K. (email: sunghwan.kim@polus-global.com) or to H.H.P. (email: xrayleox@cau.ac.kr)

Fumarate reductase is a flavoprotein enzyme containing the cofactor flavin adenine dinucleotide (FAD) that is responsible for reducing fumarate to succinate. Fumarate reductases are critical for maintaining the redox balance in the cell because they re-oxidize intracellular flavin adenine dinucleotide (FADH$_2$) or nicotinamide adenine dinucleotide (NADH) during conditions of oxygen deficiency[1] (Fig. 1a). Therefore, fumarate reductases are essential for cell survival under anaerobic conditions. Two different classes of fumarate reductases, membrane bound and soluble, have been identified. In most organisms, fumarate reductases occur as the membrane-bound form[2]. Fumarate reductases transfer electrons from iron-sulfur centers to the FAD moiety located at the active site enabling fumarate reduction[2,3]. Several bacteria, including *Shewanella*, as well as the protozoan *Trypanosoma brucei* and yeast, contain soluble fumarate reductases that can catalyze the reduction of fumarate independently from the electron transport chain[4,5]. While the membrane-bound form of fumarate reductase catalyzes a reversible reaction with a high redox potential of the covalently bound FAD, the soluble form catalyzes an irreversible reaction with a low redox potential of the non-covalently bound FAD[6,7].

The enzyme reaction mechanism of soluble fumarate reductase has been thoroughly investigated in the flavocytochrome c3 (Fcc3) family of fumarate reductases from bacteria *Shewanella*[7–9]. Fcc3 is organized into three distinct domains, the N-terminal cytochrome domain that contains four heme groups, which are critical for electron transfer, the flavin domain containing the non-covalently bound FAD and the substrate-binding active site, and the clamp domain, which is believed to be involved in controlling substrate access to the active site via structural movement. This family of enzymes uses four heme centers in the cytochrome domain to allow for appropriate electron transfer to occur.

Two soluble fumarate reductases, Frd1 and Osm1, which are essential for anaerobic growth, have been identified in *Saccharomyces cerevisiae*[10,11]. Both of these genes can be upregulated, especially under anaerobic conditions[4]. The single gene deletion of *OSM1* significantly restricts cell growth under oxygen-deficient conditions, whereas the double gene deletion of *FRD1* and *OSM1* is lethal under the same conditions, suggesting that soluble fumarate reductases are critical for survival under anaerobic conditions[4,12]. Interestingly, a sequence comparison within the Fcc3 family has suggested that both Osm1 and Frd1 contain only a flavin domain and a clamp domain, and thus lack the cytochrome domain, which is critical for electron transfer (Fig. 1b). These findings indicate that Osm1 and Frd1 might function differently than the remainder of the Fcc3 family of fumarate reductase, possibly by utilizing a distinct electron transport mechanism.

Osm1 was considered a mitochondrial protein since it was first identified[12]. However, a recent study has indicated that Osm1 can also be localized in the endoplasmic reticulum (ER). Williams et al.[13] have shown that two different types of Osm1, an ER-resident form and a mitochondria-resident form, can be produced depending on the differential use of two different translation start sites. The Osm1 protein translated from the first start codon is targeted to the ER, whereas the Osm1 protein translated from the second codon (Met33) is targeted to the mitochondria (Fig. 1b). Genetic studies have also suggested that Osm1 functionally co-operates with an ER-resident enzyme, Ero1[13–15].

In the current study, we provide two structures for Osm1: one in a complex with its cofactor (FAD) and substrate (fumarate), and another in a complex with its cofactor FAD, substrate, and a second unexpected FAD molecule located in a newly defined second FAD-binding pocket. Based on structural, biochemical, and biophysical analyses, we elucidate the molecular mechanism of fumarate reduction via electron transfer without the presence of any known electron transfer mediators such as heme or iron-sulfur centers and showed that Osm1 plays a role as a master redox regulator by replenishing oxidized flavin molecules in both the ER and the mitochondria.

## Results

**Overall structure of Osm1.** During the purification of Osm1 for this structural study, we realized that its stability was dependent on the presence of the FAD cofactor in the Osm1-cofactor-binding site (Supplementary Fig. 1). As a result, we successfully obtained sufficient holo-Osm1 by including additional FAD during the purification steps. As the purified intact Osm1 protein did not crystallize, we performed limited proteolysis with trypsin, removing flexible fragments. The trypsin-resistant Osm1 was successfully crystallized. Following an intensive effort to overcome these purification and crystallization issues, we finally solved the 1.8 Å crystal structure of Osm1. The structure was refined to a $R_{work} = 17.9\%$ and $R_{free} = 21.3\%$. The data collection and refinement statistics are summarized in Supplementary Table 1. The asymmetric unit contained one molecule, with the final model encompassing residues 32–501. Residue A31 is an extra residue arising from the cloning construct, indicating that limited trypsin digestion specifically cleaved the N-terminal 6× His-tag by recognizing the arginine in the thrombin recognition sequence. The Osm1 structure contained the FAD cofactor and the fumarate substrate in the active site (Fig. 1c).

The high-resolution structure of Osm1, in the presence of its cofactor and substrate, revealed that it comprises 17 helices, α1–α17, and 19 β-sheets, β1–β19, forming two distinct domains, a flavin domain (residues 32–268 and 386–501) and a clamp domain (residues 269–385) (Fig. 1c and Supplementary Fig. 2). The FAD- and substrate-bound active site is located in the center of the protein at the interface of the two domains, each lining part of the active site. Previous biochemical and structural studies have shown that soluble fumarate reductase in solution can exist as a monomer, a dimer, or even higher oligomeric forms[16]. To confirm the stoichiometry of Osm1 in solution, we conducted size-exclusion chromatography-multi-angle light scattering (SEC-MALS). FAD and the substrate-bound Osm1 were eluted at around 16–18 mL, which corresponds to a molecular size of ~50 kDa, indicating that it exists as a monomer in solution. This result was confirmed by MALS. The theoretical calculated molecular weight of monomeric Osm1 without any tag, FAD, or substrate is 51.15 kDa, and the experimental molecular weight from MALS was 49.8 kDa (1.04% fitting error), with a polydispersity of 1.0 (Fig. 1d). Based on our analysis using SEC-MALS, we conclude that FAD- and substrate-bound Osm1 exists as a monomer in solution.

To compare the Osm1 structure with other similar structures, the current Osm1 structure was deposited in the DALI server[17] and structurally related proteins were identified (Supplementary Table 2). The top five matches, which had Z-scores of 51.7 to 37.6, were (in order) Fcc3 from *Shewanella frigidimarina* (1QJD), Fcc3 from *Shewanella putrefaciens* (1D4C), the open form of Fcc3 from *S. frigidimarina* (1QO8), 3-ketosteroid dehydrogenese from *Rhodococcus jostii* (4AT2), and fumarate oxidoreductase from *Escherichia coli* (3P4S), indicating that Fcc3 from *S. frigidimarina* is the most structurally similar to Osm1. Among the top five matches, the top three were from the Fcc3 family and could be easily superimposed on top of Osm1 (Supplementary Fig. 3). The overall Osm1 structure was similar to that of the flavin and clamp domain structures of the Fcc3 family. The root mean square deviation (RMSD) between Osm1 and the representative Fcc3 from *S. frigidimarina* (1QJD) is 1.60 Å. Compared to the structures of other prokaryotic soluble fumarate reductases from

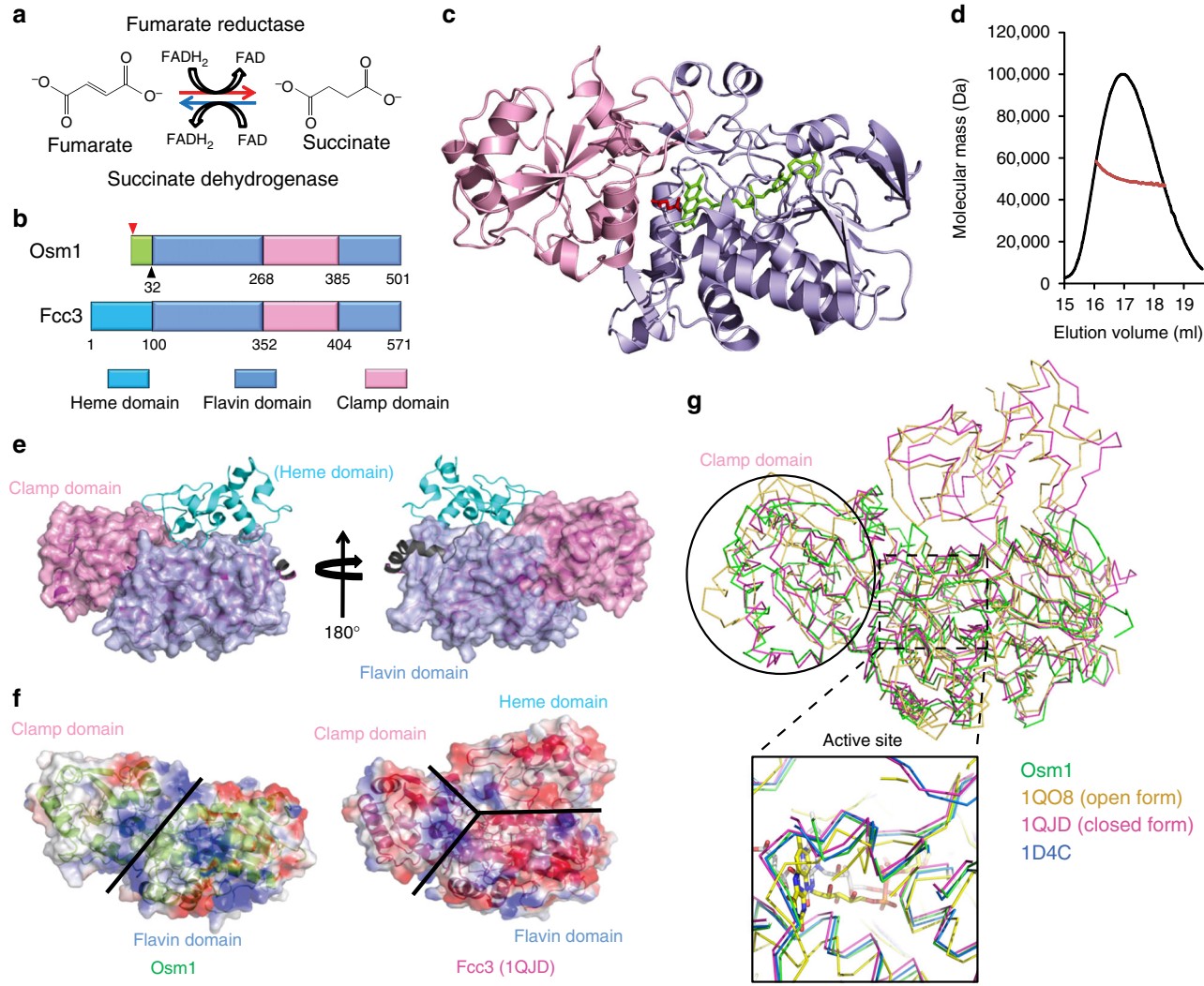

**Fig. 1** Crystal structure of Osm1 in a complex with the FAD cofactor and substrate. **a** Schematic of the enzymatic reaction catalyzed by fumarate reductase. **b** Domain structure of Osm1 and a representative soluble fumarate reductase, Fcc3, from *Shewanella frigidimarina*. The individual domains are indicated by different colors, as shown. The Osm1 construct containing residues 32–501 used in the current structural study is indicated by the black triangles. The two different translational start sites, M1 and M33, are indicated by the arrows. ER endoplasmic reticulum, Mt mitochondria. **c** Ribbon diagram of the structure of Osm1 in a complex with FAD and substrate. Two distinct domains are shown with light blue for the flavin domain and light pink for the clamp domain. Bound FAD and substrate are shown in green and red, respectively. **d** Multi-angle light scattering (MALS) measurement of Osm1 in a complex with FAD and substrate. The *x*-axis and *y*-axis indicate the elution volume and molecular mass, respectively. **e** Comparison of the active site and the FAD-binding site of Osm1 with other Fcc3 structures. The color codes are indicated. **f** Comparison of the Osm1 structure with a representative structural homolog by superposition of Osm1 with Fcc3 from *S. frigidimarina*. The clamp domain and the flavin domain are indicated in light blue and pink, respectively. The heme domain found only in the Fcc3 family is indicated in green. **g** Electrostatic surface representation of both Osm1 and Fcc3 from *S. frigidimarina*. The domain boundaries are indicated by black lines

the Fcc3 family, the most notable feature of the Osm1 structure was that it did not contain a heme domain, which is known to be critical for electron transfer (Fig. 1e). In addition, the overall features of the Osm1 electrostatic surfaces differed from those of the other Fcc3 family members. The surface of Osm1 contains a mixture of hydrophobic and charged residues, whereas the surface of Fcc3 was mainly acidic (Fig. 1f). The surface of the clamp domain in Osm1 was hydrophobic, whereas that of Fcc3 was acidic. These different features are understandable because the sequence identity between the two functionally related proteins is <25%. A solvent channel-like structure, which contained positive charges on its surface, was observed to be present between the flavin and clamp domains (Fig. 1f).

Two different forms of the Fcc3 family have been reported, namely closed and open[7,9,16]. In the Fcc3 family of fumarate

reductases, in the absence of substrate, the clamp domain is in the open form. The flexible clamp domain can close in the presence of a substrate. In such a case, the substrate helps to close the clamp domain by interacting with both the flavin domain and the clamp domain. However, it is not clear if this structural alteration, which occurs as a result of substrate binding, is critical in controlling enzyme activity[18]. Pair-wise structural alignments between Osm1 and the closed or open forms of the Fcc3 family show that the overall structure of Osm1 is more similar to the closed form of the Fcc3 family; the RMSDs between Osm1 and the open form (1QO8) and closed form (1QJD) being 4.0 and 1.6, respectively (Fig. 1g) (Supplementary Fig. 3). The clamp domain of the open form of Fcc3 did not superimpose well on Osm1. In contrast, the FAD cofactor and substrate-binding sites were almost identical to those in the Fcc3 family. The active site of the

open form of Fcc3 is not well organized when compared with that of the closed form, with the location of the FAD cofactor near the active site being slightly off-set from the typical FAD-binding site in the closed form of Fcc3 and Osm1 (Fig. 1g).

**Structural analysis of the active site**. As expected from the primary sequence (Supplementary Fig. 4), the structure of Osm1 was found to consist of two distinct domains, a flavin domain (residues 29–268 and 386–501) that holds the non-covalently bound FAD cofactor, and a clamp domain (residues 269–385) known to be involved in controlling substrate access to the active site in other soluble fumarate reductases. Although the sequence identity is relatively low across family members (e.g. ~25% identity with other soluble fumarate reductases from *Shewanella*), the functionally critical residues surrounding the active site and the FAD-binding site were all found to be conserved. The active site of Osm1 can be inferred from the presence of a fumarate molecule in the structure (Fig. 2a). The substrate- and FAD-bound active site in Osm1 is located in a cleft between the flavin domain and clamp domain. Similar to other soluble fumarate reductases, FAD is not covalently attached to Osm1. Fifteen amino acid residues, including A45, K66, N73, S74, S79, G80, V194, D260, E467, S482, and L483, from the flavin domain form hydrogen bonds of 3.5 Å or less with the FAD cofactor (Fig. 2b). Fumarate is located immediately next to the isoalloxazine ring of the FAD cofactor, having a direct interaction with the protein (Fig. 2c). S78, S79, H281, R326, H435, R477, and G480 from Osm1 are involved in the binding of fumarate in the active site through the formation of hydrogen bonds (Fig. 2d).

Previous biochemical and structural analyses have probed the active site, and consequently there is a proposed catalytic mechanism for the Fcc3 family of fumarate reductase[8,19,20]. These studies have suggested that H365, R402, H504, and R544 in Fcc3 from *S. frigidimarina* participate in substrate binding and fumarate reduction (Fig. 2e). For proton transfer, E378, R381, and R402 have all been suggested as being critical residues[20]. Interestingly, all six residues (H281, E301, R304, R326, H435, and R477) that were identified as being critical residues for the activity of Fcc3 family members[7,9] are completely conserved in Osm1 (Fig. 2e). To confirm that these candidate residues are critical for the catalytic activity of Osm1, we individually mutated all of these conserved residues to alanine. The fumarate reductase activity of Osm1 and these mutants were assessed by measuring the oxidation of free FAD in the presence of fumarate (Fig. 2f). We could induce an anaerobic environment by depleting oxygen with sodium dithionite and using air-tightening reaction plates for enzymatic assays. Under this condition, an excess amount of dithionite removes all oxygen molecules from solution as well as reduces FAD to $FADH_2$. Upon fumarate being reduced by Osm1, the $FADH_2$ molecule is oxidized to FAD, increasing its absorbance at 450 nm. Therefore, the progress of the enzymatic reaction can be monitored by measuring at $A_{450}$ (Fig. 2g). As a result, the reaction rates of the wild type (WT) and mutants of Osm1 were obtained and are compared in Fig. 2h. All of the catalytic mutants, designed based on previous studies of the Fcc3 family, had significantly reduced catalytic activity, confirming that the enzymatic mechanism is very similar to that of the Fcc3 family.

**Identification of a second FAD-binding site**. Based on the observation that Osm1 can oxidize free flavin molecules in solution (Fig. 2f), we hypothesized that Osm1 might possess an extra FAD-binding site that either allows for electron transfer between free FAD and bound cofactor FAD or through which cofactor FAD can be replaced by free FAD, since FAD is non-

covalently bound to Osm1. In order to test the first hypothesis, we added an excess amount of FAD right before crystallization. The crystal we obtained diffracted to 1.75 Å and the structure was solved by molecular replacement. The structure was refined to an $R_{work} = 18.8\%$ and $R_{free} = 23.9\%$. The data collection and refinement statistics are summarized in Supplementary Table 1. This structure showed the presence of an additional FAD molecule that could be detected right above the active site and in between the flavin domain and clamp domain (Fig. 3a). The shape of the electron density map appeared to be similar to flavin mononucleotide (FMN), showing a clear isoalloxazine ring and one phosphate group (Fig. 3b). However, because we added excessive FAD during the crystallization step, this density must have arisen due to the presence of FAD rather than FMN. The adenine nucleotide moiety of FAD, which was not anchored in this second flavin-binding pocket, could not be detected in the electron density map due to its flexibility. The second FAD-binding pocket was well defined, possessing a deep cavity containing enough space to accommodate the isoalloxazine ring from FAD (Fig. 3c and Supplementary Fig. 5). With respect to the catalytic mechanism, FAD bound in the second FAD-binding pocket may directly transfer electrons to cofactor FAD because of the lack of a barrier between the two FAD molecules. Moreover, the edge-to-edge distance between cofactor FAD and the second FAD molecule was around 4 Å, sufficiently close enough for direct electron transfer (Fig. 3d).

There are three known cellular flavin cofactors, namely riboflavin, FMN, and FAD. The electron density map that appeared to correspond to FMN, rather than FAD, suggested that other riboflavin cofactors could also be oxidized by Osm1 in solution. In order to test this hypothesis, all three riboflavin molecules were assessed for their ability to be oxidized by Osm1 in solution. As a result, all three riboflavin molecules were found to be well oxidized by Osm1 under anaerobic conditions with only minor differences in Km value (Fig. 3e and Supplementary Table 3). However, Osm1 could not oxidize or bind non-flavin electron carrier molecules, such as phenazine methosulfate, indicating that the second FAD-binding pocket is specific to flavin structure (Supplementary Fig. 6). FAD showed at least a 2-fold lower Km value (2.858 μM) than the other flavin molecules (5.847 and 9.375 μM), indicating that FAD might be the preferred riboflavin molecule among them (Supplementary Table 3).

The second FAD-binding pocket was composed mostly of hydrophobic residues. Six amino acid residues, including W295, F297, L298, N359, F362, and Y363, from the clamp domain and three amino acid residues, including K76, S78, and P162, from the flavin domain of Osm1 formed the binding pocket of the isoalloxazine ring from FAD (Fig. 3f, g). These nine residues are located within 3.5 Å of FAD. Among these nine residues, P162, W295, F297, F362, and Y363 directly interact with FAD via hydrophobic interactions, whereas L298 forms a hydrogen bond with FAD using a nitrogen from the main chain (Fig. 3g). Given this second FAD-binding pocket, we investigated whether FAD binding to this pocket is critical for Osm1's ability to oxidize free $FADH_2$ in solution. To assess this, we mutated two amino acid residues, S78K and P162R, which are located in the second FAD-binding pocket that appeared to be important in accommodating riboflavin molecules. Compared to WT Osm1, the S78/P162R double mutant showed a much lower ability to oxidize free $FADH_2$ in solution (Fig. 3h). To compare the FAD-binding ability of Osm1 and the S78/P162R double mutant, we performed isothermal calorimetry (ITC). Injection of FAD into the ITC sample cell containing holo-Osm1 created an exothermic response, confirming that their interaction is comprised of hydrogen bonding and hydrophobic interactions as indicated by the favorable binding enthalpy (Δ*H*, −22.7 kJ/mol) and entropy

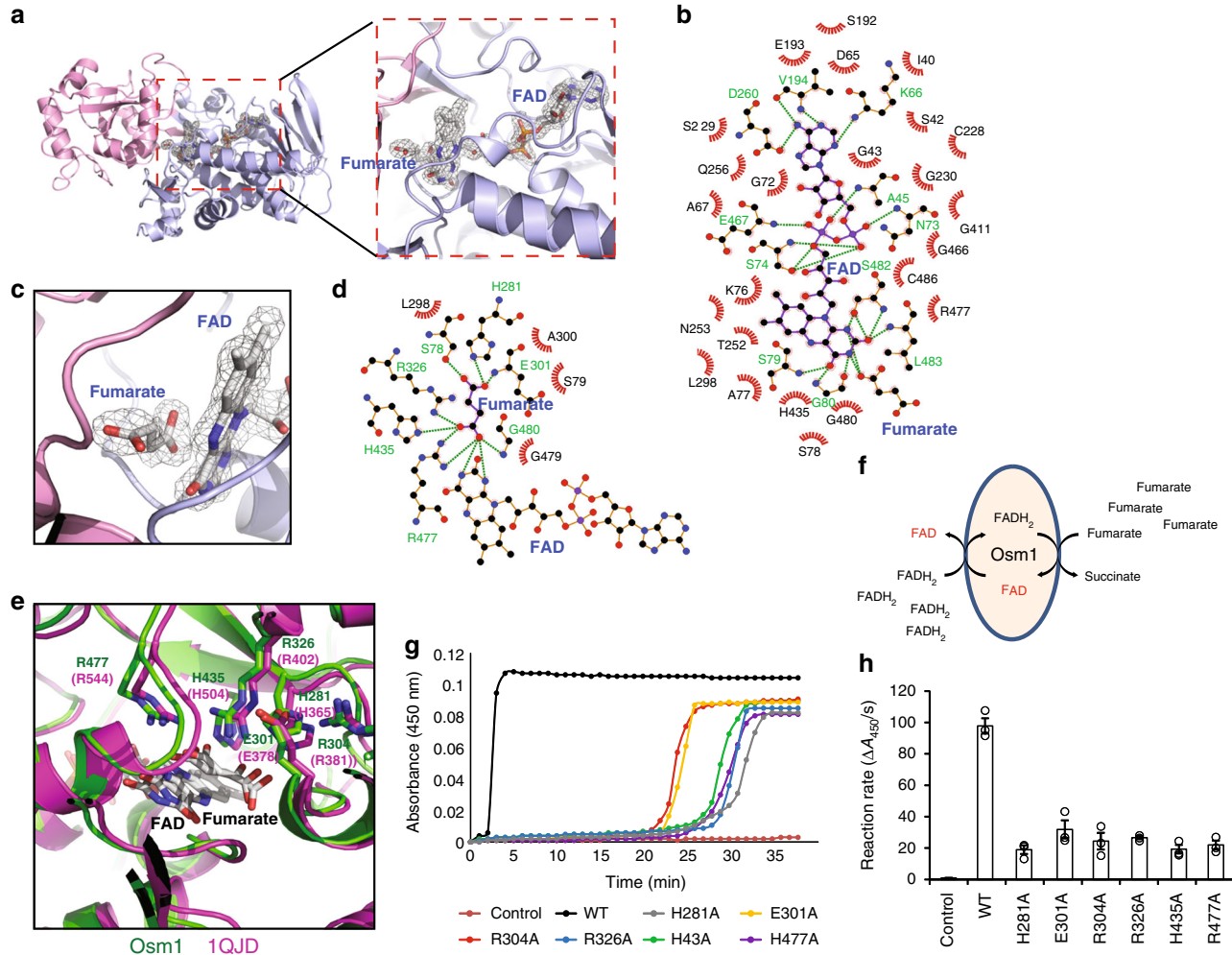

**Fig. 2** Structural analysis of the catalytic site in Osm1. **a** An omit density map contoured at the $1-\sigma$ level around the active site. The flavin and clamp domains are shown in blue and pink, respectively. The active site is shown by the red dotted rectangle. The FAD and substrate-binding site is magnified for better visualization. **b** A 2D FAD-Osm1 interaction diagram. The residues involved in the FAD interaction are indicated. Green-colored residues are involved in hydrogen bonding. The green dotted lines indicate hydrogen bonds. The 2D interaction diagram was generated by Ligplot[30]. **c** The substrate-binding site, magnified to show a clear electron density map. **d** The 2D substrate (fumarate)-Osm1 interaction diagram. The residues involved in the substrate accommodation are indicated. **e** Structural comparison of the active site of Osm1 with Fcc3 fumarate reductase from *Shewanella frigidimarina* (1QJD). The location of six completely conserved amino acids which may be critical for Osm1 activity, H281 (H365), E301 (E378), R304 (R381), R326 (R402), H435 (H504), and R477 (R544), are indicated. **f** Electrons can be transferred from FADH$_2$ to fumarate in the absence of oxygen. The fumarate reductase activity of Osm1 can be assessed by monitoring the oxidation of free FADH$_2$ based on the absorbance of FAD at 450 nm. **g** Comparison of fumarate reductase activities of Osm1 wild type and its catalytic mutants. Oxidation of FADH$_2$ was monitored under anaerobic environment induced by sodium dithionite. A representative reaction curve among three replications was provided. **h** Bar graph of fumarate reductase activity of Osm1 wild type and its mutants. The corresponding dot plots were overlaid in the bar charts. The reaction rates were calculated based on reaction curves ($n = 3$, error bar was derived by standard error)

factor ($T\Delta S$, 7.1 kJ/mol) with an affinity (Kd) of 4.92 μM (Fig. 3i, k). However, injection of FAD into the S78K/P162R mutant did not show any detectable thermal response, indicating low or no interaction (Fig. 3j, k). Taken together, we confirmed that Osm1 can oxidize free riboflavin molecules by accommodating them in the second FAD-binding pocket, which allows electron transfer between cofactor FAD and free FADH$_2$.

**Anaerobic disulfide bond formation by Osm1 and Ero1.** Ero1 is an essential ER-resident flavoenzyme that generates de novo disulfide bonds by reducing the FAD cofactor. Under aerobic conditions, FADH$_2$ can be easily re-oxidized by transferring electrons to a di-oxygen molecule (Fig. 4a). However, it is still not clear how the FAD cofactor in Ero1 is re-oxidized in the absence

of oxygen (Fig. 4a). In this study, we found that a fraction of Osm1 was water insoluble and glycosylated, suggesting its residency in the ER membrane (Supplementary Fig. 7). The ER residency of Osm1 and its genetic interaction with Ero1 were also suggested by previous studies[13,15,21]. Because our structures showed that Osm1 has the ability to oxidize free flavin molecules under anaerobic conditions and that Osm1 does not contain a heme domain, it may be possible that Osm1 interacts with other proteins that can transfer electrons. Therefore, we hypothesized that the FAD cofactor in Ero1 is re-oxidized by transferring electrons to fumarate via the FAD cofactor of Osm1 under anaerobic conditions. Based on their co-localization in the ER membrane, we first examined if Ero1 can directly interact with Osm1 by performing an immunoprecipitation of yeast cell lysates over-expressing both Ero1 and Osm1, as well as a pull-down

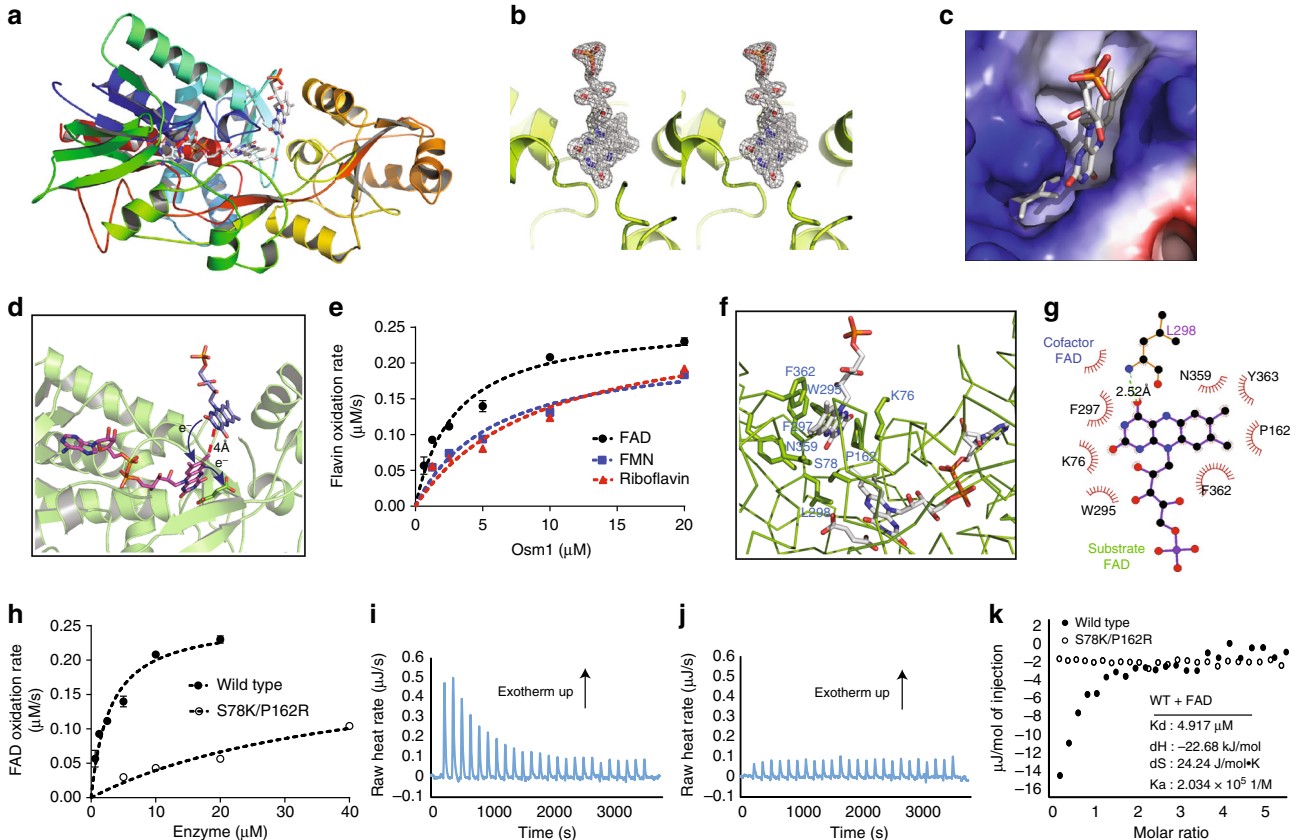

**Fig. 3** Osm1 crystal structure revealing the second FAD-binding pocket. **a** Ribbon diagram of the structure of a second FAD bound to Osm1. The chain from the N- to C-termini is colored blue to red. **b** A stereo image of an omit density map contoured at the 1−σ level around the second FAD-binding site. **c** Electrostatic surface in the second FAD-binding pocket. **d** Active site of Osm1 including a second FAD-binding site. A model of the second FAD-mediated electron transfer from the second FAD to substrate (fumarate) via the FAD cofactor (first FAD) is indicated. **e** All of the reduced FAD, FMN, or riboflavin (50 μM) were re-oxidized by Osm1 in the presence of fumarate. **f** Details of the environment of the second FAD-binding site. The residues involved in the FAD interaction are indicated. **g** A 2D diagram of the interaction between the second FAD and Osm1. The residues involved in the second FAD interaction are indicated. **h** WT Osm1 (closed circle) and S78K/P162R mutant (open circle) were incubated with reduced free FAD in the presence of fumarate. The mutant Osm1 oxidized FAD more slowly than WT Osm1. **i**, **j** Free FAD binding to WT Osm1 (**i**) and S78K/P162R (**j**) were analyzed by ITC. **k** The integrated heat area of WT Osm1 (closed circle) and mutant (open circle) were plotted by FAD-to-Osm1 molar ratio, calculating the binding affinity between WT Osm1 and free FAD. The experiments for calculating catalytic reaction rates were performed three times for **e**, **h**, and error bars were derived by standard deviation

experiment with bacterial lysates over-expressing both MBP-tagged Ero1 and 6× His-tagged Osm1. Immunoprecipitation of FLAG-tagged Ero1 with anti-FLAG antibody beads showed that there was weak binding with Osm1-HA, but there was stronger binding than negative control with Ero1-3 × Myc (Fig. 4b). A sequential pull-down experiment showed a weak but significant interaction between Ero1 and Osm1 (Fig. 4c). To validate this weak interaction, we measured their interaction in vitro using surface plasmon resonance (SPR). A range of different concentrations of purified Osm1 were injected onto a CM5 SPR chip immobilized with purified Ero1, resulting in a concentration-dependent response increase in binding to Ero1 with a Kd value of around 200 μM based on the saturation curve (Fig. 4d). The interaction was confirmed by ITC. Injection of 1.6 mM Osm1 into 40 μM Ero1 in the ITC sample cell created an exothermic response with a Kd of ~150 μM, which is similar with affinity value from SPR (Fig. 4e). To analyze the position of the interaction, we conducted a docking simulation using the known crystal structures of Ero1 (PDB ID: 3ahq) and the currently solved crystal structure of Osm1. The docking simulation suggested that Ero1 bound to Osm1 near the heme domain of Fcc3 (Fig. 4f). To validate this tentative interaction between Ero1 and

Osm1, we designed a chimeric Osm1 containing a heme domain from Fcc3 at its N-terminus, to mimic the Fcc3 family (Fig. 4g). Although the heme domain in the chimeric Osm1 was misfolded, which was determined by spectroscopy and sodium dodecyl sulfate-polyacrylamide gel electrophoresis (SDS-PAGE) analyses (Supplementary Figure 8A and B), the Fcc3-Osm1 chimera did not bind Ero1 in the SPR assay (Fig. 4h), indicating that the presence of a misfolded heme domain prevents a physical interaction between Ero1 and Osm1 at the tentative Ero1-binding site of Osm1 proposed by docking analysis and structure of another soluble fumarate reductase family (Supplementary Figure 8C). The Fcc3-Osm1 chimera still possessed full fumarate reductase activity (Fig. 4i). Taken together, these findings demonstrate that Ero1 can mimic the heme domain by directly interacting with Osm1 to transfer electrons to the FAD cofactor of Osm1. This interaction might therefore be another way for anaerobic disulfide bond formation in the ER by transferring electron from Ero1-cofactor FADH₂ to fumarate via the Osm1-cofactor FAD.

In order to re-constitute anaerobic disulfide bond formation by Osm1 and Ero1, we combined thioredoxin oxidation by Ero1 and fumarate reduction by Osm1 (Fig. 4j). Using a tightly sealed tube, Ero1, Osm1, and reduced Trx1 were mixed together, and an

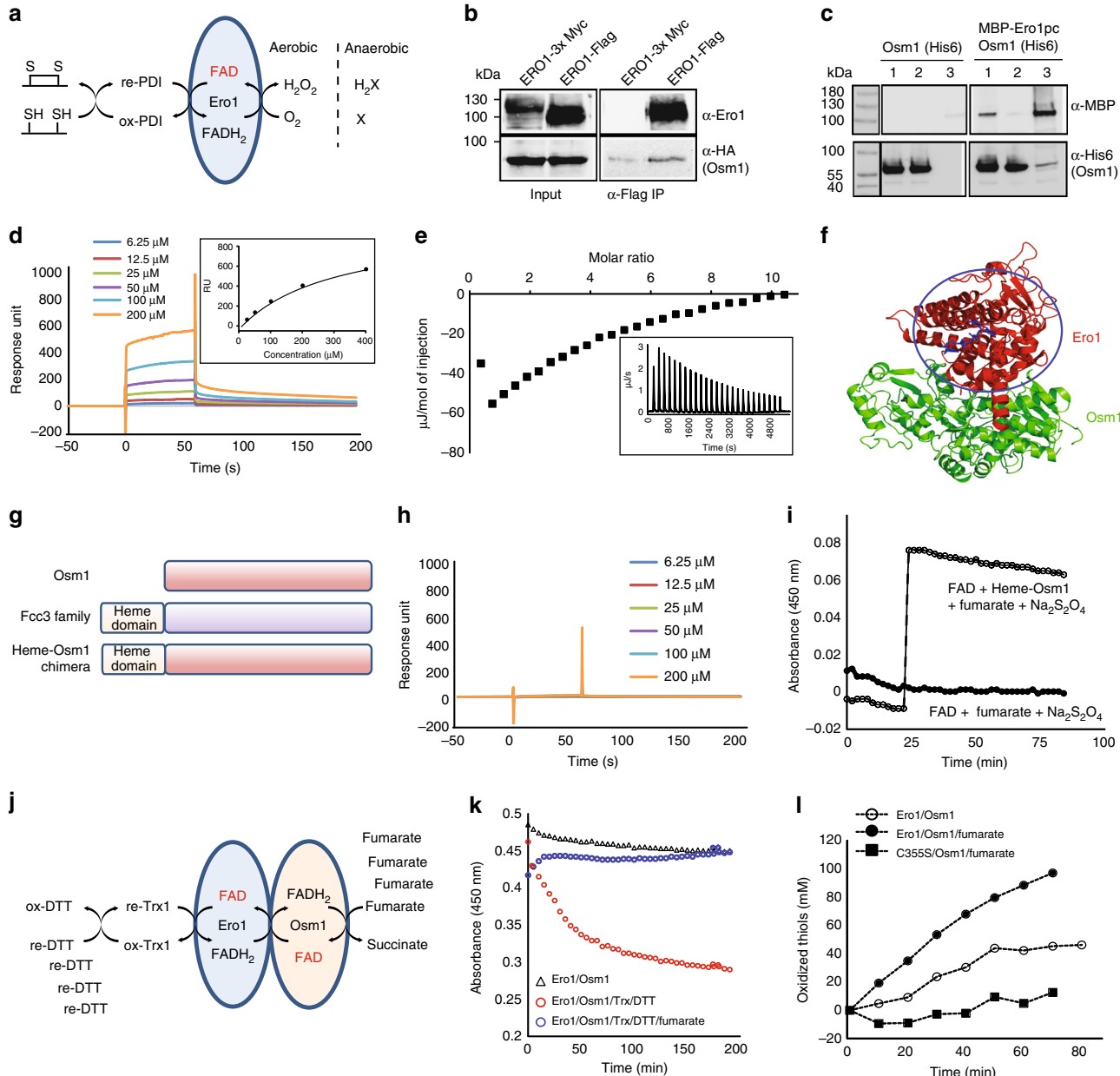

**Fig. 4** Physical interaction between Osm1 and Ero1. **a** Ero1 function. **b** Co-immunoprecipitation study. Cell lysates from yeast cells co-expressing Ero1-3×Myc or Ero1-FLAG were immunoprecipitated using anti-FLAG antibody. Samples were separated by SDS-PAGE and immunoblotted using anti-Ero1 antibody or anti-Osm1 serum. **c** In vitro pull-down assay. The interaction between Osm1 and Ero1 was analyzed by sequential pulldown with nickel then amylose beads. Samples were separated by SDS-PAGE and immunoblotted with anti-MBP and anti-6His antibodies. Lane 1: input; lane 2: fraction not bound to amylose beads; lane 3: 20× fraction bound to amylose beads. **d** Surface plasmon resonance. Purified MBP-Ero1 was immobilized and Osm1 was then injected over the surface. All sensorgrams were processed by subtracting the value of a blank sensorgram. Equilibrated response unit plotted against Osm1 concentration is shown in the black box. **e** The interaction between Osm1 and Ero1 confirmed by ITC. Integrated heat area of Ero1 was plotted by Osm1-to-Ero1 molar ratio. **f** Molecular docking of Osm1 with Ero1. **g** Domain composition of the Fcc3-Osm1 chimera. **h** Surface plasmon resonance. The purified Fcc3-Osm1 chimera was immobilized and Osm1 was then injected. **i** Reductase activity assay for the Fcc3-Osm1 chimera. Oxidation of FAD by 0.5 μM Fcc3-Osm1 chimera measured based on absorbance at 450 nm. **j** Model system used to assess Osm1-Ero1-mediated thiol oxidation. **k, l** In an airtight environment, when oxygen was depleted by overreaction between Ero1 and a reduced thiol mix (thioredoxin/DTT), the Ero1- and Osm1-bound FAD was reduced. However, fumarate inhibited this FAD reduction by working as an electron acceptor (**k**). At the same time, oxidized thiol molecules in reaction increased (**l**). The catalytically inactive Ero1 mutant, C355S, could not oxidize thiols even in the presence of Osm1 and fumarate

anaerobic-like environment was created by adding a high concentration of dithiothreitol (DTT) to deplete dissolved oxygen. In the absence of fumarate, the absorbance at 450 nm due to the enzyme-bound FAD cofactor decreased as oxygen was consumed to oxidize DTT (Fig. 4k, open circle); however, fumarate could maintain the enzyme-bound FAD cofactor in its

oxidized form (Fig. 4k, closed circle) similar to the level of the mixture without reduced thiols, thioredoxin, or DTT (Fig. 4k, open triangle). At the same time, we also assessed disulfide bond formation by measuring the amount of reduced thiols using 5,5-dithiobis(2-nitrobenzoic acid) (DTNB). As expected, in the presence of fumarate, Ero1 and Osm1 could generate disulfide

bonds in an anaerobic-like environment (Fig. 4l). We found that the catalytic activity of Ero1 was critical for anaerobic disulfide bond formation by showing that the Ero1 catalytic mutant, C355S, could not oxidize thiols (Fig. 4l, closed square)[22,23].

**Inhibition of Osm1 by blocking the second FAD-binding site.** The above data suggest that electrons can be transferred from Ero1 to fumarate by Osm1 for de novo disulfide bond formation, which occurred even under anaerobic conditions. Next, we analyzed whether disulfide bond formation could be prevented by blocking the Ero1-Osm1-coupled electron transfer system. Since flavin molecules are capable of binding to the second FAD-binding site, chemical compounds containing a flavin-like moiety, such as Cibacron Blue 3G-A, could be potential pocket binders (Fig. 5a). Cibacron Blue 3G-A has been widely used for the protein purification of reductases, including yeast-soluble fumarate reductases[24]. However, we could not exclude the possibility that chemical compounds containing a flavin-like moiety might function as oxidizing agents for Ero1. Therefore, we first tested if Ero1 could use such other flavin-containing molecules as an oxidizing source under the anaerobic-like environment used in the previous enzyme assays (Fig. 4j). In these experiments, excess flavin molecules were supplied instead of Osm1 and fumarate, and the reduction of the flavin molecules was analyzed (Fig. 5b). Interestingly, Ero1 could only use free FAD, but not FMN or riboflavin, under oxygen-depleted conditions as Ero1-oxidized thiol mix (Fig. 5c–e). We further confirmed that a catalytic mutant, Ero1-C355S, could not reduce FAD. These results indicate that Ero1 can transfer electrons only to FAD, whereas Osm1 can accept electrons from all flavin molecules via the second FAD-binding site. This difference between Ero1 and Osm1 suggests that the compounds containing a flavin-like moiety prevent disulfide bond formation by blocking the second FAD-binding pocket. Cibacron Blue 3G has an anthraquinone group, which is similar to the flavin ring (Fig. 5a). A docking simulation suggested that the anthraquinone region of Cibacron Blue 3G could bind to the flavin ring-binding pocket (Fig. 5f). The sulfonic acid functional groups in Cibacron Blue 3G had tight interactions with hydrogen bond donor residues, including K76, S78, and K255 in Osm1. As a result, Cibacron Blue 3G inhibited the disulfide bond formation catalyzed by the Ero1-Osm1-coupled system (Fig. 5g). This observation supports the hypothesis that blocking the second FAD-binding pocket with a small molecule can prevent electron transfer between Osm1 and Ero1 under anaerobic conditions (Fig. 5h).

## Discussion

*Saccharomyces cerevisiae* expresses Frd1 and Osm1, which are both soluble fumarate reductases. Despite the critical roles played by soluble fumarate reductases for survival in yeast under anaerobic conditions, their molecular mechanism has not yet been clearly elucidated. Frd1 is localized in soluble form in the cytosol, whereas as discussed and studied previously, Osm1 can be localized in both mitochondria and the ER, where an oxidizing environment is important for functions of these organelles. Moreover, ER-resident Osm1 may be as important as mitochondria-resident Osm1. ER is an important organelle for the expression and oxidative folding of secretory proteins; therefore, the ER must maintain an oxidizing environment even under anaerobic conditions.

Yeast-soluble fumarate reductase has evolved by the removal of a heme-binding domain compared to bacterial-soluble fumarate reductase. Instead of the heme-binding domain, Osm1 appears to have reserved a site for the direct binding of free flavin molecules as well as other flavoenzymes. Several lines of evidence in support

of this hypothesis were found in this study. First, the structure we resolved revealed the presence of a second FAD-binding site in Osm1, and our biochemical experiments suggest that yeast-soluble fumarate reductases can oxidize free flavin molecules such as FAD, FMN, and riboflavin to create an oxidizing environment under anaerobic conditions. This provides a way for the cell to maintain an oxidizing environment, especially in the mitochondria and ER where many oxidases have to function properly in the absence of oxygen (Fig. 6a).

Interestingly, a second FAD-binding pocket circumstance composed of several aromatic residues in Osm1, which could explain the lower affinity of the second FAD and the function of Osm1, was used to obtain pai–pai stacking interactions with the only isoalloxazine ring of FAD. This interaction strategy suggests that the second FAD-binding site could serve as a relatively low-affinity site to allow for an efficient exchange with residual reduced flavin molecules, while the first FAD-binding site could provide a high-affinity site to restrict the exchange of FAD. Second, biochemical and biophysical experiments confirmed the physical interaction between a flavoenzyme and Osm1. Through this interaction, these two enzymes could transfer electrons more efficiently (Fig. 6b). In either case, the second FAD-binding site was critical for the function of Osm1. For example, an ER-resident flavoenzyme, Ero1, which generates de novo disulfide bonds and transfers them to protein disulfide isomerase for oxidative protein folding of secretory proteins in the ER (Fig. 6b), must transfer electrons to oxidants. Considering that free FAD can serve as an oxidant to obtain electrons from Ero1 in the absence of oxygen, the oxidation of free $FADH_2$ by Osm1 via the second FAD-binding pocket must be sufficient to allow for anaerobic disulfide bond formation.

Our biophysical studies showed that Ero1 directly binds Osm1 in place of the heme domain. Although their interaction was not of high affinity (Kd = ~200 μM), these proteins might readily interact, because both Ero1 and Osm1 are associated with the ER membrane, resulting in a high local concentration of each protein. Direct interaction-mediated electron transfer between two FAD molecules in Osm1 and Ero1 is still controversial because they are more than 20 Å apart in the docking model, which is too far for direct electron transfer. With this information, we speculated and proposed two possible electron transfer mechanisms. Model 1: Structural changes of Ero1-mediated close location of two FAD molecules. It has been known that the human ortholog of Ero1 undergoes conformational changes upon activation[25]. In addition, yeast Ero1 is activated by reducing its three regulatory disulfide bonds in the presence of a substrate[23]. Therefore, it might be possible that yeast Ero1 also undergoes conformational changes upon activation or binding with Osm1, leading to two FAD molecules close enough for electron transfer. Model 2: Indirect electron transfer via amino acid residues. We may have missed other components needed to mediate indirect electron transfer system between Ero1 and Osm1, such as specific amino acid residues that are involved in electron transfer proposed by a previous soluble fumarate reductase study[20]. Since the structural information of either active yeast Ero1 or the Ero1/Osm1 complex are not available, it is hard to determine the exact mechanism of Osm1-Ero1 electron transfer. However, the two models proposed derived from previously characterized Ero1 and Osm1 may be the best way to describe this electron transfer system.

In support of the association between these proteins, we observed that an Osm1-heme chimeric protein lost its binding ability to Ero1, but could still mediate thiol oxidation with Ero1 in the absence of oxygen. This result indicated that the physical interaction between Ero1 and Osm1 was not necessary, but was important for efficient and stable electron transfer. For example,

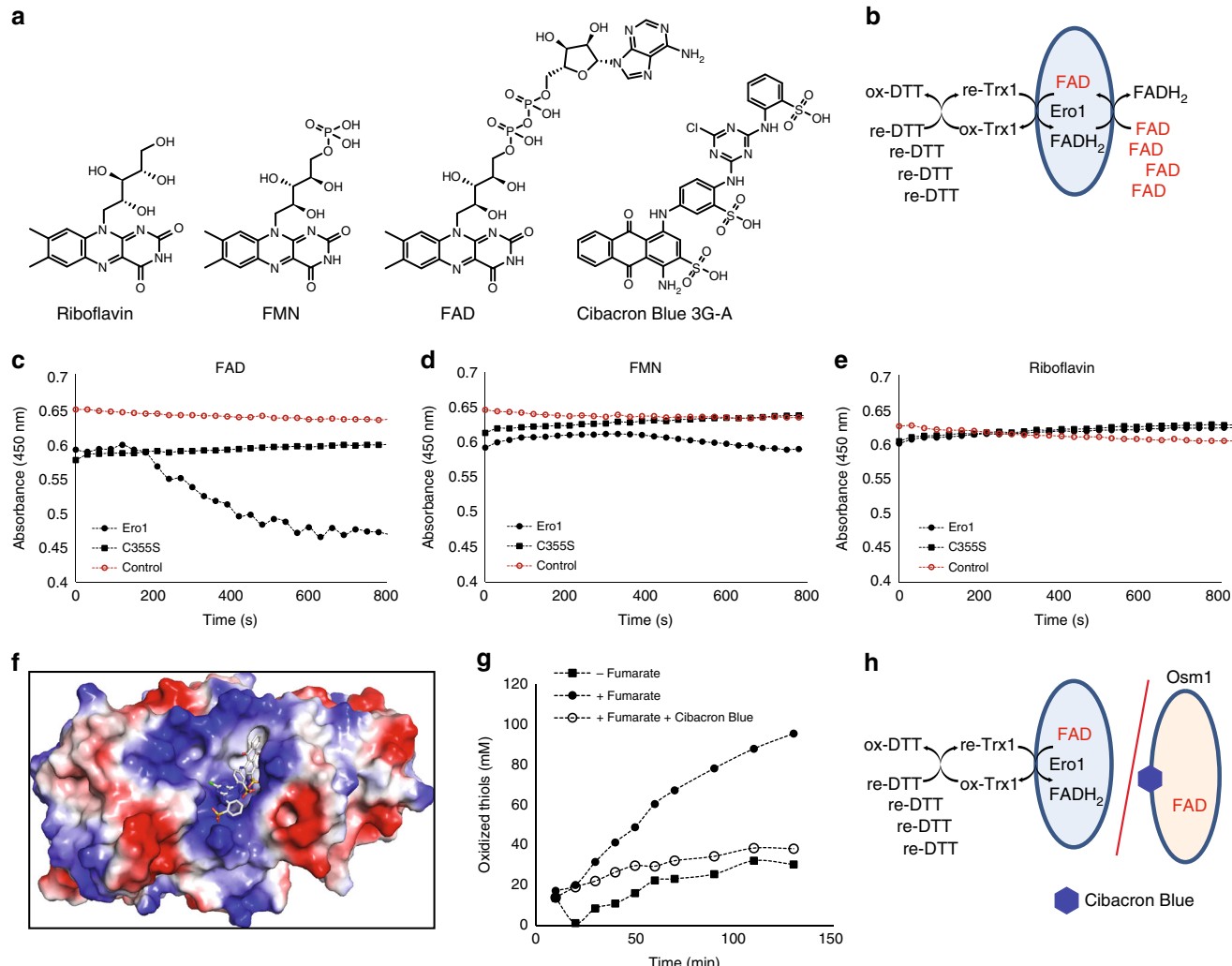

**Fig. 5** Effect of blocking the second FAD-binding pocket. **a** Chemical structures of riboflavin, FMN, FAD, and Cibacron Blue 3G-A. **b** Model system used to assess the ability of Ero1 to transfer electrons to free FAD. **c–e** Ero1 can only transfer electrons to FAD. Under airtight conditions, Ero1, or Ero1-C355S, was activated by adding a reduced thiol mix (DTT and thioredoxin) along with FAD (**c**), FMN (**d**), and riboflavin (**e**). Reduction of the flavin molecules was then monitored by changes in absorbance at 450 nm. **f** Docking simulation for Cibcron Blue 3G-A binding to Osm1. **g** Cibacron Blue 3G-A inhibits anaerobic thiol oxidation by the Ero1-Osm1 system. **h** Model system used to assess inhibition of disulfide bond formation by the Ero1-Osm1 system

Osm1 or Ero1 may lose their FAD cofactors in the absence of their binding partner, since FAD associates with them non-covalently. FAD cofactor-deficient recombinant Osm1 (apo-Osm1) has a much lower stability than holo-Osm1 (Supplementary Fig. 1), so their physical interaction increases the stability of the enzyme itself and therefore the efficiency of electron transfer. To enable the Osm1-mediated catalytic cycle for oxidation of either free FADH₂ or reduced Ero1, binding of another fumarate molecule or efficient fumarate exchange is necessary. Since the second fumarate-binding site was not detected in the structural study, it is more reasonable that fumarate is efficiently exchanged in the cell. When the free fumarate concentration is higher than Osm1, fumarate can be exchanged since fumarate-binding affinity for substrate-binding site is relatively low (mM scale based on enzymatic assay).

Altogether, based on our current structural and biochemical studies, we have shown that soluble fumarate reductase, Osm1, in eukaryotes can maintain an oxidizing environment under anaerobic conditions, either by oxidizing cellular flavin cofactors or by direct interaction with other flavoenzymes. These data suggest that Osm1 plays the role of a master redox regulator by replenishing oxidized flavin molecules in both the ER and mitochondria.

## Methods

**Protein expression and purification**. The *S. cerevisiae* OSM1 gene sequence corresponding to amino acids 32–501 was cloned in pET28a vector using *Nhe*I and *Xho*I restriction sites. The point mutants (H281A, E301A, R304A, R326A, H435A, R477A, and S78/P162R) were generated by site-directed mutagenesis. Plasmids were transformed into BL21 (DE3) *E. coli*-competent cells and their expression was then induced by treating the bacteria with 0.25 mM isopropyl β-D-thiogalacto-pyranoside (IPTG) for 25 h at 18 °C. Cells were harvested by centrifugation and sonicated in re-suspension buffer containing 20 mM NaPO₄, pH 7.4, 500 mM NaCl, 25 mM imidazol, and a protease inhibitor cocktail. Cell lysates were clarified by centrifugation at 10,000 *g* for 1 h and filtration with a 0.22 μm filter. The protein, which contained an N-terminal His-tag and a thrombin recognition sequence, was purified by nickel affinity followed by gel-filtration chromatography. A Superdex 200 gel-filtration column 10/30 (GE Healthcare) that had been pre-equilibrated with a solution of 20 mM Tris-HCl, pH 8.0, and 150 mM NaCl was used for gel-filtration chromatography. The protein eluted at around 15–16 mL via gel-filtration chromatography was collected and concentrated to 9–10 mg/mL for crystallization and further experiments.

To generate chimeric Osm1 containing the N-terminal heme-binding domain, a codon-optimized DNA corresponding to the heme-binding domain of *Shewanella* Fcc (A13-P135) was inserted into pET28a-Osm1. Chimeric Osm1 was purified by nickel affinity, anion exchange (HiTrap Q, GE Healthcare), and gel-filtration

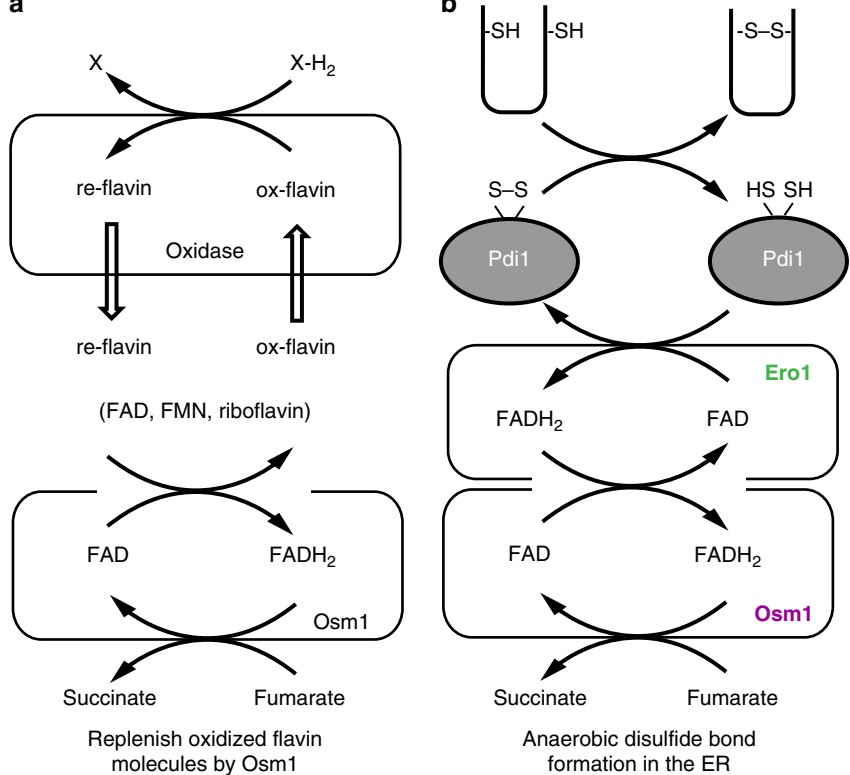

**Fig. 6** Model of the function of Osm1. Our proposed models showed that Osm1 functions as a master regulator of the redox state of flavin molecules (**a**) and disulfide bond formation in the ER under anaerobic conditions (**b**)

chromatography (Superdex 200 column) using a column that had been pre-equilibrated with phosphate-buffered saline (PBS) buffer. For obtaining recombinant Ero1 and its catalytic mutant C355S proteins, $His_6$-maltose-binding protein ($H_6$-MBP)-fused Ero1 (56–424) and Ero1-C355S were expressed in Origami B(DE3)-competent cells by induction with 0.4 mM IPTG supplemented with 10 μM FAD. $H_6$-MBP-fused protein was purified from cell lysates by affinity to HiTrap $Ni^{2+}$ column (GE Healthcare) and Amylose (New Engeland Biolabs Inc.). $H_6$-MBP was cleaved off by tobacco etch virus (TEV) protease (1:100 wt/wt protease to substrate) for 15 h at 25 °C. Ero1 and Ero1-C355S were collected by flowing through the HiTrap $Ni^{2+}$ column (GE Healthcare) to remove uncleaved fusion protein, cleaved $His_6$-tagged MBP, and TEV protease.

**Multi-angle light scattering**. The absolute molar mass of Osm1 was determined by MALS. Briefly, the target protein was loaded onto a Superdex 200 HR 10/30 gel-filtration column (GE Healthcare) that had been pre-equilibrated in buffer containing 20 mM Tris-HCl, pH 8.0, and 150 mM NaCl. The ÄKTA chromatography system (GE Healthcare) was coupled to a MALS detector (mini-DAWN Treos) and a refractive index detector (Optilab DSP) (both Wyatt Technology).

**Thermal stability assay**. The protein stability of Osm1 was determined by measuring melting curves with protein thermal shift dye (Invitrogen) in StepO-nePlus Real-Time PCR (Thermo Fisher Scientific), as described in the manufacturer's instructions. Briefly, 2 μg Osm1 was mixed with 1× protein thermal shift dye and 20 μl buffer and signal changes upon protein denaturation were monitored with a temperature increase from 40 to 80 °C.

**Structure determination and analysis**. Native diffraction data sets were collected on beamline SBII-5C of the Pohang Accelerator Laboratory (PAL), Republic of Korea, at a wavelength of 0.9776 Å. The data sets were indexed and processed using HKL-2000. The structure was determined by the molecular replacement phasing method using Phaser[26]. The cytochrome domain-deleted flavocytochrome C3 structure from *S. frigidimaria* (PDB code: 1QJD)[9], which shared 25% sequence identity (32% after removing the cytochrome domain) with Osm1, was used as a search model. Model building and refinement were conducted by COOT[27] and Refmac5[28], respectively. Water molecules were added using the ARP/wARP function in Refmac5. The geometry was inspected using PROCHECK and was found to be acceptable. A total of 97.65% of the amino acids were located in the most favorable region, whereas 2.35% were in the allowed regions of the Rama-chandran plot. In the case of the second FAD-bound Osm1 structure, 97.44 and

2.56% of the amino acids were located in the most favorable region and in the allowed region, respectively. All molecular figures were generated using the Pymol program[29].

**Fumarate reductase assay**. For this assay, 4 μM purified Osm1 and its mutants, including chimeric Osm1, were mixed with 100 mM fumarate (Sigma) and 50 μM FAD (Sigma) in PBS with 1 mM EDTA in either 96- or 384-well plates. FAD was reduced and anaerobic conditions were generated by adding sodium dithionite (Sigma) to give a final concentration of 0.5% and then sealing the plates from air. Oxidation of FAD by Osm1 was monitored by measuring the absorbance at 450 nm. The maximum reaction rate was calculated using Synergy 4 (BioTek).

**Pull-down assay and immunoprecipitation**. Ero1 (56–424) fused to MBP and Osm1 were inserted into pCDFDuet and pET28a vectors, respectively, containing different origins of replication. These two recombinant vectors were co-transformed into BL21(DE3) cells and then selected for using spectinomycin and kanamycin. Lysates from *E. coli* over-expressing both MBP-Ero1 and His-tagged Osm1 were prepared by inducing their expression with IPTG and homogenizing them by sonication. After removing the cell debris, the Ero1-Osm1 interaction was tested by sequential pull-down experiments using Ni-NTA and amylose beads. First, the soluble lysates were incubated with Ni-NTA beads, after which they were washed with five column volumes of HEPES-buffered saline (HBS) containing 10 mM imidazole and eluted with 200 mM imidazole. Protein mixtures pulled down by Ni-NTA were then loaded onto the amylose beads and washed with HBS, followed by elution with 10 mM maltose. Samples from each step were analyzed by SDS-PAGE and immunoblotted using anti-His (1:1000, Abcam_ab18184) and anti-MBP (1:1000, Thermo_PA1-989) antibodies and infrared-conjugated secondary antibodies (1:10,000 Li-COR_P/N 925-32211 or P/N 925-68025).

For immunoprecipitation from yeast cell lysates, FLAG-tagged or 3× Myc-tagged *ERO1*-coding *URA3* 2μ plasmids were co-transformed with 3× HA-tagged *OSM1*-coding *CEN LEU2* plasmids into BY4742. Cells were grown in minimum media supplemented with uracil and leucine, and lysates were prepared by disrupting the cells with acid-washed beads. Cell lysates were incubated in Tris-buffered saline (TBS), 1% Triton, and a protease inhibitor cocktail (Roche) for 1 h on ice. Clarified cell lysates were mixed with anti-FLAG affinity beads (Sigma) for 1 h on ice and then washed with TBS and 1% Triton. Eluted samples were analyzed by an anti-Ero1 serum which was obtained by injection of Ero1p-c protein into rabbits by Covance Inc. (Denver, PA, USA) or an anti-HA antibody (1:1000, Santa Cruz Biotechnology_sc-7392) in conjunction with the appropriate secondary

antibodies (1:10,000). All uncropped scans of the blots are provided at Supplementary Figure 9.

**SPR and ITC**. SPR experiments were performed using a BIACORE T-200 (GE Healthcare) equipped with a Series CM5 sensor chip. MBP-Ero1 (83 kDa, > 90% pure based on SDS-PAGE) was immobilized using amine-coupling chemistry as indicated in the BIACORE T-200 wizard program. MBP-Ero1 at a concentration of 30 μg/mL in 10 mM sodium acetate, pH 5.0, was immobilized at a density of 1500RU. A reference cell was prepared by blocking with ethanolamine. To analyze the binding affinity to Ero1, Osm1 or chimeric Osm1 in PBS was injected over the two flow cells at concentrations ranging from 6.25 to 200 μM at a flow rate of 30 μL/min at 25 °C. The Biacore T-200 evaluation software (GE Healthcare) was used for data processing and the Kd value was calculated by affinity fitting.

For affinity analysis between Osm1 and FAD or Ero1, isothermal titration calorimetry was used. For the interaction between Osm1 and Ero1, 2.5 μL of 1.6 mM Osm1 was injected into a nano-ITC cell (TA Instrument) containing 40 μM Ero1 in PBS. For the interaction between free FAD and Osm1, 2.5μL of 1 mM FAD was injected into nano-ITC cell containing 30 μM holo-Osm1 or 28 μM S78/P162R double mutant.

**Thiol oxidation assay**. As Ero1 oxidizes thiols (thioredoxin/DTT) in an airtight environment, oxygen is depleted and Ero1-bound FAD cannot be re-oxidized, losing its absorbance at 450 nm or fluorescence at 450/520 nm; this process is called bleaching (Fig. 4j). Therefore, Osm1-mediated thiol oxidation in the absence of oxygen could be assessed either by monitoring FAD bleaching or measuring residual reduced thiols with DTNB. Recombinant Osm1 and Ero1 (or the Ero1-C355S mutant) were mixed to a volume of 10 μM, with or without 20 μM reduced Trx, 100 mM DTT, and 100 mM fumarate in PBS containing 1 mM EDTA. To evaluate the Cibacron 3B inhibitory effect, the Osm1 activity was determined as described above, except that Cibacron 3B 100 μM was added.

**Docking simulation**. Molecular docking for the Ero1-Osm1 interaction was performed using the Autodock 4.2 package. The Ero1 structure was prepared from the Protein Data Bank ID 3ahq. Cibacron Blue 3G docking to Osm1 was studied using the Maestro 2016-4 (Schrödinger Suite) software. The second FAD-bound Osm1 X-ray crystal structure was prepared by removing all water and hydrogen assignments at pH 7.0 with the Protein Preparation Wizard module. Cibacron Blue 3G was minimized by using the conjugate gradient algorithm and the OPLS2005 force field with Minimization module in Maestro. The receptor grid was generated and ligand docking was accomplished with the Glide module.

## Data availability

Atomic coordinates and structure factors have been deposited in the Protein Data Bank with PDB ID: 5GLG (Osm1) and 5ZYN (second FAD-bound Osm1). A reporting summary for this Article is available as a Supplementary Information file. All other data supporting the findings of this study are available from the corresponding authors on reasonable request.

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

## Acknowledgements

We thank the staff at BL-5C of the PAL (Pohang, Korea) for their kind help with data collection. This study was supported by the Basic Science Research Program through the National Research Foundation of Korea (NRF) of the Ministry of Education, Science, and Technology (NRF-2014R1A1A1003451, NRF-2017M3A9D8062960, and NRF-2018R1A2B2003635) and a grant from the Korea Healthcare Technology R&D Project, Ministry of Health & Welfare, Republic of Korea (HI17C0155).

## Author contributions

H.H.P. and S.K. designed and supervised the project. Y.-J.S., R.K.S., and J.Y.C. performed cloning, expression, and protein purification. J.Y.C. and C.M.K. crystallized and collected X-ray data. J.Y.C., C.M.K., and H.H.P. solved the protein structure. T.-H.J. performed MALS. Y.-J.S. and S.K. performed the enzyme activity assays. Y.-J.S., J.S., and H.K. performed the pull-down assay. Y.-J.S., C.M.K., and S.K. performed SPR and the ITC

experiments. Y.L. performed the docking simulation. H.H.P., C.A.K., and S.K. wrote the manuscript. All the authors have discussed the results, commented on the manuscript, and approved the manuscript.

## Additional information

**Competing interests:** The authors declare no competing interests.

