## [Peer Review File · Nature Communications]

Reviewers' comments:

Reviewer #1 (Remarks to the Author):

In this work, the authors solved crystal structures of a fumarate reductase from yeast, Osm1, in two forms; one in complex with FAD and fumarate and the other in complex with two molecules of FAD. The authors also performed a series of biochemical and biophysical analyses to elucidate molecular mechanisms of electron relays between Osm1 and excess FAD and between Osm1 and a flavoenzyme Ero1, a likely physiological redox partner of Osm1. How protein disulfide bonds are formed and how the redox homeostasis is maintained in the ER of eukaryotic cells under anaerobic conditions remain an important open question. This work tackles the important problems.

Overall, the experiments have been designed well and seem to have been done appropriately in order to draw confirmative conclusions. Regrettably, however, there are a number of unclear sentences, redundant phrases and grammatical errors in the text. Precise descriptions are not provided especially regarding the procedure and interpretation of experimental data. There are also much to be improved in figures. As a result, the current version of manuscript is hard to understand for broad readers of this journal and hence could not be acceptable for publication without major revisions described below.

Major points:

1) Fig. 1C is not well displayed. Bound FAD and fumarate molecules should be highlighted so that readers can easily recognize their binding sites in Osm1. There is no need that the ribbon diagram is colored from blue to red. A flavin domain and clamp domain should be shown in different colors separately as in Figure 2A.

2) While the authors state by citing Figure 1G that the overall structure of Osm1 is more similar to the closed form of the Fcc3 family than to the open form, the figure does not support it at all. The superposition of the overall structures of Osm1, and the open and closed forms of the Fcc3 family should be displayed in Figure 1.

3) As to the second FAD binding site shown in Figure 3F and 3G, several aromatic amino acids seem to locate to make tight π - π stacking interactions with the isoalloxazine ring of FAD. Meanwhile, this site should serve as a relatively low-affinity site to allow the efficient exchange with residual FADH₂. In contrast, the first FAD binding site likely disallows the exchange with free FAD due to the high affinity. In terms of the different modes of interaction between FAD and its neighboring residues at the first and second binding sites, this issue should be discussed.

4) To enable the Osm1-mediated catalytic cycle for oxidation of free FADH₂ or reduced Ero1, binding of another fumarate molecule or efficient fumarate exchange is necessary. In this context, this reviewer wonders whether or not the authors tried to crystallize Osm1 in the presence of excess fumarate molecules. Even though the second fumarate binding site has not been identified, the authors should discuss how reduced form of bound fumarate (i.e. bound succinate) is converted to or replaced with fumarate.

5) As to the ITC analysis shown in Fig. 3I-3K, why did not the authors titrate 'FADH₂' to Osm1. Taking into account the real enzymatic reaction catalyzed by Osm1, the authors should analyze the binding of FADH₂ to Osm1 rather than that of FAD. Besides, the horizontal axis of the ITC data shown in Fig. 6K should be relabeled with the FAD-to-Osm1 molar ratio, instead of injection number, so that the readers can readily see the stoichiometry of free FAD to holo-Osm1.

6) Based on the docking model of the Ero1-Osm1 binary complex shown in Fig. 4F, Ero1 seems to compete with FADH₂ for binding to Osm1. Meanwhile, Ero1 itself can be oxidized by excess free FAD to significant extent (Fig. 5C). After all, which is more likely under physiological settings, direct oxidation of Ero1 by Osm1 or oxidation of Ero1-released FADH₂ by Osm1? Does the addition

of free FAD enhance the Osm1 catalysis of Ero1 oxidation? What is the physiological level of free FAD in the ER? With the additional experimental data and information raised above, the authors should discuss this issue.

7) It is unclear why the binding of Ero1 to Osm1 was analyzed by SPR, not by ITC, although ITC could provide more detailed information on thermodynamic parameters for protein-protein interactions, in general. The reason should be mentioned.

Minor points:

1) page 6, near the top; The sentence beginning with "Because the purified" does not seem to make sense. From the observation that Osm1 subjected to limited proteolysis was successfully crystallized, this conclusion could be drawn.

2) page 6, near the bottom; "from the other Fcc3" should read "from those of the other Fcc3".

3) page 8, near the top; The sentence beginning with "Under these conditions" does not seem to make sense. Although it is understandable that dissolved oxygen is quenched with added dithionite, it is unclear what actually reduces FAD to FADH₂ in this situation?

4) page 9, near the bottom; It is unclear why Ero1 was expected to bind to Osm1 near the heme domain of Fcc3 before the inhibitory role of the heme domain in the functional interaction between Ero1 and Osm1 is shown by biochemical and biophysical experiments. The phrase "as expected" should be deleted.

5) page 10, middle; the verb "asked" should be replaced with a more appropriate verb.

6) The horizontal axis of the ITC data shown in Fig. 6K should be shown by molar ratio, not by injection number, which facilitates us to see the stoichiometry of free FAD to holo-Osm1.

Reviewer #2 (Remarks to the Author):

This paper describes structure and associated solution data for a soluble fumarate reductase from yeast. These enzymes are very similar in sequence to previously studied bacterial soluble fumarate reductases, the main difference being the latter have a 4 heme domain which is lacking in the yeast enzymes. The authors reveal the structure of the yeast enzyme, and demonstrate binding of a 2nd flavin in close proximity of the FAD cofactor. They provide solution data that leads them to propose a model for physiological function: to link fumarate as alternative terminal electron acceptor to systems requiring oxidation under anaerobic conditions.

I think this is an interesting paper, with considerable merit. However, I have some reservations:

1) the paper is overly long, and on many occasions repeats itself. The introduction almost becomes a discussion before the data is shown. The Fig legends are good, but a lot of this text is also repeated in the main text. Some Figures are redundant too. As such, the manuscript would be much better if it would be reduced in size by 1/3 at least.

2) The 2nd flavin binding is interesting, especially since there seem to be very little polar contacts established, which suggests this might be a more generic binding pocket. Have the authors used other (non-physiological) electron acceptors (i.e. DCPIP, PMS?). The solution data regarding the various flavin forms is not convincing in my opinion. The errors suggest there is not a big difference. Second, these are shown with far too many significant numbers. The only solid conclusion I can see is that flavins (i.e. FMN/FAD/riboflavin) are all oxidized.

3) The kinetic data displayed in Fig 2G is not very useful. There is a distinct lag phase which is due to presumably consumption of excess dithionite.... Hence, any observed rates will depend on enzyme activity with dithionite, enzyme activity with flavin, (slow) oxygen re-entry into the system etc. This should be done more carefully, so that reproducible values of the various mutants can be shown. Given the fact that the entire active site is conserved with bacterial enzymes, the observation these mutations hinder activity is not that surprising.

4) The authors go on to use Ero1 as a substrate, rather than free flavins. This appears to work, and the authors claim Ero1 might dock at the position of the heme domains (as observed in the bacterial systems). The authors use a hybrid yeast enzyme that contains a fusion to a bacterial heme domain. There is little characterisation of this latter fusion enzyme. Are the hemes incorporated? If so, does this enzyme perform heme reduction. If not? Is the heme domain folded/present?

5) The nature of the proposed direct Osm to Ero electron transfer is not entirely clear, would the bound FAD from Osm be within 14 Å of the Ero FAD in their docking model? If not, how do the authors propose electrons are transferred?

6) There is mention of another paper for the crystallisation/purification details. This in my opinion should be provided for a manuscript to a journal of this impact level.

Reviewer #3 (Remarks to the Author):

Kim et al. describe a first crystal structure of a eukaryotic soluble fumarate reductase at 1.8 Å resolution. The structure was solved by molecular replacement using the 32% identical cytochrome c3 structure from *Shewanella frigidimarina* (after removal of its cytochrome domain) as a search model. Not surprisingly, large parts of the Osm1 structure, including the flavin (residues 32-268 and 386-501) and the clamp (residues 269-385) domains already known from previously determined structures of bacterial fumarate reductases. The only new aspect is the identification of an additional flavin cofactor binding site (close enough to the first one for direct electron transfer) in a second crystal structure at 1.75 Å resolution based on co-crystallization of Osm1 with excess FAD. Although the authors do not address this directly, Supplementary Table 1 appears to indicate that this second FAD molecule or, more precisely the FMN portion of it that could be modelled is associated with much higher B-factors. In addition, this partial modelling of FMN would indicate that the affinity of FMN to Osm1 should be higher than that of FAD, which is contradicted by the findings in Supplementary Table 3.

The discussion mirrors the impression from the results section, the first half re-stating the obvious (flavin binding domain and clamp domain). If the conclusion is that the structure determined resembles the closed form, it is not understandable why the comparison in Supplementary Table 2 contains the less-closed form of *W. succinogenes* QFR rather than the more-closed form available since 2001.

The second half of the discussion is predominantly composed of words like "... seems ... appears ... suggest ... can ... could ... could ... can ... may ... might ... could ... may ... may ... may ... may ..." and therefore hardly convincing.

In conclusion, the "solid" results are essentially the direct consequence of the input (= search model for molecular replacement), all potentially novel aspects appear not well defined.

Reviewers' Comments to Author:

Reviewer: 1

We thank the reviewer for his/her constructive comments.

Comments (Major points):

1. Fig. 1C is not well displayed. Bound FAD and fumarate molecules should be highlighted so that readers can easily recognize their binding sites in Osm1. There is no need that the ribbon diagram is colored from blue to red. A flavin domain and clamp domain should be shown in different colors separately as in Figure 2A.

Response: As per reviewer's suggestion, we replaced Fig. 1C with a new image that highlights bound FAD and fumarate molecules.

2. While the authors state by citing Figure 1G that the overall structure of Osm1 is more similar to the closed form of the Fcc3 family than to the open form, the figure does not support it at all. The superposition of the overall structures of Osm1, and the open and closed forms of the Fcc3 family should be displayed in Figure 1.

Response: Thank you for finding the mislabeled figure. After careful revision, and based on the reviewer's comment, we realized that Figures 1E and 1F were also mislabeled. We corrected the sequence of Figures 1E, 1F, and 1G, and also updated Figure 1G by replacing the original image with a superposition of the overall structures of Osm1 and the open and closed forms of the Fcc3 family. As such, we now present a direct comparison between the current structure and the two different forms of bacterial fumarate reductases in Figure 1G and Supplementary figure 3.

3. As to the second FAD binding site shown in Figure 3F and 3G, several aromatic amino acids seem to locate to make tight pi-pi stacking interactions with the isoalloxazine ring of FAD. Meanwhile, this site should serve as a relatively low-affinity site to allow the efficient exchange with residual FADH₂. In contrast, the first FAD binding site likely disallows the exchange with free FAD due to the high affinity. In terms of the different modes of interaction between FAD and its neighboring residues at the first and second binding sites, this issue should be discussed.

Response: Thank you for your input on this extremely important point. We also feel that the limited interaction with lower affinity of the second FAD is critical for the function of Osm1 and have now highlighted this issue in our Discussion section.

4. To enable the Osm1-mediated catalytic cycle for oxidation of free FADH₂ or reduced Ero1, binding of another fumarate molecule or efficient fumarate exchange is necessary. In this context, this reviewer wonders whether or not the authors tried to crystallize Osm1 in the presence of excess fumarate molecules. Even though the second fumarate binding site has not been identified, the authors should discuss how reduced form of bound fumarate (i.e. bound succinate) is converted to or replaced with fumarate.

Response: We attempted to crystallize Osm1 (10 mg/ml, ~200 μ M) in the presence of excess fumarate (~10 mM) but could not find the second fumarate binding. Via an enzymatic assay with increasing fumarate concentration, we found that fumarate's binding affinity was relatively low (mM scale based on enzymatic assay, see attached unpublished raw data). Low affinity can allow fumarate (succinate after reduction) to be replaced with free fumarate, dependent on concentration. This means that the second fumarate binding site is not necessary. We discussed this in the text.

[**Response Figure 1. Osm1 oxidizes free FADH₂ with 2.8 mM K_d value.** 100 nM Osm1 was incubated with 100 μ M FAD and an increasing concentrations of fumarate. Oxygen was depleted by adding sodium dithionite in the air-tight plate. The oxidation rate of free FADH₂ was determined by monitoring the increase of FAD. As fumarate increased, oxidation of free-FADH₂ by Osm1 was faster. The K_d value between Osm1 and FAD was estimated to be ~2.8 mM based on the graph.]

5. As to the ITC analysis shown in Fig. 3I-3K, why did not the authors titrate 'FADH₂' to Osm1. Taking into account the real enzymatic reaction catalyzed by Osm1, the authors should analyze the binding of FADH₂ to Osm1 rather than that of FAD. Besides, the horizontal axis of the ITC data shown in Fig. 3K should be relabeled with the FAD-to-Osm1 molar ratio, instead of injection number, so that the readers can readily see the stoichiometry of free FAD to holo-Osm1.

Response: This is very good point. Thank you very much for this comment. We agree that we should analyze the binding of FADH₂ to Osm1 considering a possible enzymatic reaction. Unfortunately, this experiment would be too difficult to carry out, since FADH₂ can be easily oxidized in the presence of oxygen. In order to do this, we would have to keep the ITC machine in a vacuum or include an excess amount of reductant in the

experiment, such as sodium dithionite. However, high concentrations of sodium dithionite result in extreme physical properties in biochemical or biophysical experiments, such as SPR and ITC. As such, we chose to analyze binding affinity using FAD instead. That being said, it is possible that FAD and FADH₂ have different binding affinities, although we believe that binding affinity is not a matter of structures but of concentrations. Under specific conditions where FADH₂ is more abundant than FAD, Osm1 preferably binds with FADH₂ regardless of minor differences in binding affinity.

We revised Fig. 3K by relabeling with molar ratio as per the reviewer's suggestion.

6. Based on the docking model of the Ero1-Osm1 binary complex shown in Fig. 4F, Ero1 seems to compete with FADH₂ for binding to Osm1. Meanwhile, Ero1 itself can be oxidized by excess free FAD to significant extent (Fig. 5C). After all, which is more likely under physiological settings, direct oxidation of Ero1 by Osm1 or oxidation of Ero1-released FADH₂ by Osm1? Does the addition of free FAD enhance the Osm1 catalysis of Ero1 oxidation? What is the physiological level of free FAD in the ER? With the additional experimental data and information raised above, the authors should discuss this issue

Response: We believe that the reviewer's questions are really important and equally difficult to answer. According to Tu. BP 2002 Mol Cell, Ero1 activity is enhanced by free FAD under aerobic conditions. However, no one has shown that this is the same for Osm1 under anaerobic conditions. We have seen that free FAD increased Ero1 catalytic activity with Osm1 without oxygen molecules (please find below Response Figure 2, completed under anaerobic conditions). As the concentration of free FAD is increased, the Osm1-mediated Ero1 catalytic reaction gets faster. Physiologically, it has been reported that wild-type yeast cells have around 15 μ M total FAD and 3 μ M free FAD (Gliszczynska and Koziolowa, 1998). Therefore, we considered that the ER lumen also have a certain amount of free FAD. We believe that Free FAD-mediated electron transfer between Osm1 and Ero1 could be physiologically more relevant. Unfortunately, we could not include this data in our paper because it was generated by our family group a long time ago and will be published shortly.

References

1. The FAD- and O₂-Dependent Reaction Cycle of Ero1-Mediated Oxidative Protein Folding in the Endoplasmic Reticulum (2002) Mol Cell 10, 983-994
2. Gliszczynska, A., and Koziolowa, A. (1998). Chromatographic determination of flavin derivatives in baker's yeast. J. Chromatogr. A 822, 59-66.

[Response Figure 2. Free-FAD accelerated Osm1-mediated anaerobic disulfide bond formation. Under oxygen depleted conditions, 0.5 μM Ero1 and Osm1 were incubated with 250 μM reduced Trx1, 5 mM fumarate, and a concentration series of FAD, as indicated in the figure. The reaction was quenched by adding 1 mM DTNB, which reduced Trx1. The amount of reduced Trx1 was monitored with A412. A control, in which Osm1 was not included, did not show any oxidation of Trx1 even though Ero1 and 2 μM free FAD were present. As free FAD increased, anaerobic disulfide bond formation was faster]

7. It is unclear why the binding of Ero1 to Osm1 was analyzed by SPR, not by ITC, although ITC could provide more detailed information on thermodynamic parameters for protein-protein interactions, in general. The reason should be mentioned

Response: In the case of protein-protein interaction analysis, at least in our experience, the preparation and handling of experimental samples for SPR were easier than for ITC, where high concentrations of one protein (injected sample) were necessary. This is why we used SPR for this interaction study and ITC for FAD-Osm1 binding study. As per the reviewer's suggestion, we also performed ITC by preparing 1.6 mM Osm1, which is a very high concentration but was suitable in solution. The ITC value was similar with SPR, producing a K_d value of $\sim 150 \mu\text{M}$. We included this ITC experiment in Fig 4E.

Comments (Minor points):

1. Page 6, near the top; The sentence beginning with "Because the purified" does not seem to make sense. From the observation that Osm1 subjected to limited proteolysis was successfully crystallized, this conclusion could be drawn.

Response: We are sorry for providing an incorrect sentence. We modified the sentence to: "Because the purified intact Osm1 protein was unable to be crystallized, we performed limited proteolysis with trypsin, producing a trypsin-resistant Osm1 fragment"

2. Page 6, near the bottom; "from the other Fcc3" should read "from those of the other Fcc3".

Response: We have corrected this.

3. Page 8, near the top; The sentence beginning with “Under these conditions” does not seem to make sense. Although it is understandable that dissolved oxygen is quenched with added dithionite, it is unclear what actually reduces FAD to FADH₂ in this situation?

Response: Thank you for highlighting this mistake. We have revised the manuscript to explain the experimental conditions more clearly.

4. Page 9, near the bottom; It is unclear why Ero1 was expected to bind to Osm1 near the heme domain of Fcc3 before the inhibitory role of the heme domain in the functional interaction between Ero1 and Osm1 is shown by biochemical and biophysical experiments. The phrase “as expected” should be deleted.

Response: We removed “as expected” as per the reviewer’s suggestion.

5. Page 10, middle; the verb “asked” should be replaced with a more appropriate verb

Response: We replaced this with “analyzed” which we think is a more appropriate verb.

6. The horizontal axis of the ITC data shown in Fig. 6K should be shown by molar ratio, not by injection number, which facilitates us to see the stoichiometry of free FAD to holo-Osm1.

Response: We revised Fig 3K by relabeling with molar ratio as per the reviewer’s suggestion.

Reviewer: 2

We thank the reviewer for his/her positive assessment of our work.

Comments:

1. The paper is overly long, and on many occasions repeats itself. The introduction almost become a discussion before the data is shown. The Fig legends are good, but lot of this text is also repeated in the main text. Some Figures are redundant too. As such, the manuscript would be much better if it would be reduced in size by 1/3 at least.

Response: As suggested by the reviewer, we have reduced the length of the manuscript as much as possible. Notably, we have simplified all the discussions in the “Introduction” section. In addition, we removed all the repeated and unnecessary sentences in the “Results” and “Discussion” sections.

2. The 2nd flavin binding is interesting, especially since there seem to be very little polar contacts established, which suggests this might be a more generic binding pocket. Have the authors used other (non-physiological) electron acceptors (i.e. DCPIP, PMS?). The solution data regarding the various flavin forms is not convincing in my opinion. The errors suggest there is not big difference. Second, these are shown with far too many significant numbers. The only solid conclusion i can see is that flavins (i.e. FMN/FAD/riboflavin) are all oxidized.

Response: We agree with Reviewer 2’s concern. Our study shows that there is a second FAD binding site on Osm1, which is a novel and unexpected result. Our crystal structure analysis showed that there was only an FMN moiety at the second FAD binding site, despite adding excess amounts of FAD. Based on this crystal structure, which showed the importance of the FMN moiety for binding of FAD to Osm1, we also believed that other flavin molecules such as FMN and riboflavin could acts as alternative electron transfer molecules. From the enzyme reaction, as you noted, Osm1 could oxidize free FMN and riboflavin as well as FAD (Fig. 3E).

Although we did not test our hypothesis with other non-physiological electron acceptors, such as DCPIP, analysis of the crystal structure strongly suggests that a flavin moiety could come into contact with to the second FAD binding site. We have screened various chemical compounds, including a ring structure to compete riboflavin binding. Only a few compounds containing flavin-like moieties showed inhibition of Osm1-mediated FAD oxidation under anaerobic conditions. One of them, Cibacron Blue 3G, inhibited Osm1-mediated thiol-oxidation by competing with FAD at the second binding site (Fig. 5G). Moreover, docking simulation analyses showed that Cibacron blue 3G could occupy the second FAD binding site.

To provide evidence that the 2nd FAD binding site is indeed specific to a flavin moiety, we performed an Osm1-mediated anaerobic oxidation assay using non-physiological electron mediators (PMS and DCPIP) as suggested by the reviewer. PMS and DCPIP were highly reduced by sodium dithionite with loss of their specific absorbance. However, PMS and DCPIP were not re-oxidized in the presence of Osm1 and fumarate, while free FAD was re-oxidized under the same conditions (see data below). In addition, PMS

failed to interact with Osm1 in the ITC experiment. These data indicate that the 2nd FAD binding site is specific to flavin molecules. We included this data at Supplementary Figure 6.

[Response Figure 3 (Supplementary Figure 6). Binding analysis of non-flavin electron carrier molecules to Osm1. Anaerobic oxidation of electron carrier molecules, phenazine methosulfate (PMS) and 2,6-dichlorophenolindophenol (DCPIP) by Osm1 and fumarate have been compared to that of FAD. All electron carrier molecules, 100 µM, were reduced by sodium dithionite, then incubated with 4 µM Osm1 in air-tight sealed plates. The oxidation process was monitored over time at appropriate wavelengths specific to each molecule. In contrast to FAD, neither PMS nor DCPIP were re-oxidized by Osm1. Additionally, PMS did not show binding to Osm1 in the isothermal titration calorimetry experiment, indicating that the second FAD binding site is specific to flavin molecules.

3. The kinetic data displayed in Fig 2G is not very useful. There is a distinct lag phase which is due to presumably consumption of excess dithionite.... Hence, any observed rates will depend on enzyme activity with dithionite, enzyme activity with flavin, (slow) oxygen re-entry into the system etc. This should be done more carefully, so that reproducible values of the various mutants can be shown. Given the fact that the entire active site is conserved with bacterial enzymes, the observation these mutations hinder activity is not that surprising.

Response: As far as we understand from the reviewer's comments, this enzyme kinetic assay was not useful because the experimental results were too obvious since the selected mutants were well-conserved within the Fcc family (previously analyzed with Fcc family). Regarding the reviewer's concerns that the enzymatic assay conditions were too

harsh to obtain reproducible results, we have now included data to show that well-known key residues studied in the prokaryotic system were also catalytically important in eukaryotic system.

We realize that an experiment containing high amounts of sodium dithionite is not the best way to determine Osm1 activity. However, anaerobic enzyme reactions and sodium dithionite are widely used to deplete oxygen and mimic anaerobic conditions. As noted by the reviewer, the lag phase indicates the depletion of sodium dithionite. However, the length of the lag phase is dependent on enzyme activity to free FADH₂. As noted by the reviewer, it is indeed possible for oxygen to re-enter the reaction. However, Fig. 2G indicates that a control (dark red) without enzymes showed no oxidation of FAD for a certain period of time (35 min). This control data suggests that 35 minutes is long enough to conduct a safe kinetic assay, since the amount of re-entered oxygen was not high enough to produce detectable levels of FAD.

In order to develop this assay, we tested various concentrations of sodium dithionite and found the that best condition shows a detectable reaction for Osm1 wild type as well as all Osm1 mutants. We carried out the experiment in triplicate and calculated the average of the reaction rates, indicating the error bars (Fig 2H). Using the best experimental conditions with a reasonable control and a repeated assay, we repeated this experiment to obtain reproducible data.

4. The authors go on to use Ero1 as a substrate, rather than free flavins. This appears to work, and the authors claim Ero1 might dock at the position of the heme domains (as observed in the bacterial systems). The authors use a hybrid yeast enzyme that contains a fusion to a bacterial heme domain. There is little characterization of this latter fusion enzyme. Are the hemes incorporated? If so, does this enzyme perform heme reduction. If not? Is the heme domain folded/present?

Response: As per the reviewer's suggestion, we checked whether the hemes are incorporated in the forcibly fused heme-binding domain. According to our spectroscopic analysis, the chimera was not able to produce a peak at 402 nM, whereas heme-containing P450 (control) produced a peak at 402 nM, indicating that our chimera did not contain heme (Figure A).

During the revision process, we solved the chimeric-Osm1 structure. Strangely, the heme-domain was not present in the structure (three Osm1 molecules in ASU). Based on this unexpected structure and the reviewer's comments on the chimera protein, we analyzed the fusion protein and found that a forcibly fused heme domain does not fold well and is subsequently degraded out from Osm1 after ~5 days (Figure B). Because all of the chimeric assay was performed within 5 days, we believe that the misfolded heme domain at the N-terminus of chimeric Osm1 inhibited the binding of Ero1. Although the heme domain was misfolded, it blocked the binding of Ero1 at the tentative Ero1 binding site of Osm1, as proposed by docking analysis and the structure of another soluble fumarate reductase family, the Fcc3 family (Figure C). Based on the results of this misfolded-heme analysis, we modified the explanation of the chimera study in the manuscript and included our analysis in Supplementary Figure 8.

[Response: Figure 4 (Supplementary Figure 8). Characterization of chimeric Osm1 containing a heme domain. A) Absorbance scan of Osm1 chimera to detect heme. P450 containing heme in the protein was used as a positive control. Osm1 that does not contain heme was used as negative control. The chimera did not exhibit a peak at 402 nm (blue line), indicating that the chimera did not contain heme in the fused heme domain. **B)** Osm1 chimera proteins were subjected to SDS-PAGE and degradation patterns in the time scale were analyzed. **C)** Proposed chimera structure model. Misfolded fused heme domain inhibits interaction with Ero1.]

5. The nature of the proposed direct Osm to Ero electron transfer is not entirely clear, would the bound FAD from Osm be within 14 Å of the Ero FAD in their docking model? If not, how do the authors propose electrons are transferred?

Response: Based on the reviewer's and editor's comment, we attempted to explain the proposed Osm1-Ero1 electron transfer mechanism in greater detail with this limited docking information. As noted by the reviewer, we measured the distance between the two FAD molecules and realized that they were more than 20 Å apart, which is too far for direct electron transfer. With this information, we speculated and proposed two possible electron transfer mechanisms.

Model 1) Structural changes of Ero1 mediated by the close proximity of two FAD molecules. It is known that the human orthologue of Ero1 undergoes conformational changes upon activation (ref1). In addition, yeast Ero1 is activated by reducing its three regulatory disulfide bonds in the presence of a substrate (ref2). Therefore, it might be

possible that yeast Ero1 also undergoes conformational changes upon activation or binding with Osm1, resulting in two FAD molecules that are close enough for electron transfer. Model 2) Indirect electron transfer via amino acid residues. We may have not taken into account other components involved in mediating the indirect electron transfer system between Ero1 and Osm1, such as the specific amino acid residues involved in electron transfer proposed by a previous soluble fumarate reductase study (ref3).

Since the structural information of neither active yeast Ero1 nor Ero1/Osm1 complex are available, it is hard to determine the exact mechanism of Osm1-Ero1 electron transfer. The two proposed models, derived from previously characterized Ero1 and Osm1, may be the best way to describe this electron transfer system.

We proposed these models in the manuscript with proper citations (see Discussion).

References

1. Inaba K, Masui S, Iida H, Vavassori S, Sitia R, Suzuki M. (2010). Crystal structures of human Ero1 α reveal the mechanisms of regulated and targeted oxidation of PDI. *EMBO J.* 29 (19), 3330-3343
2. Sevier CS, Qu H, Heldman N, Gross E, Fass D, Kaiser CA (2007). Modulation of cellular disulfide-bond formation and the ER redox environment by feedback regulation of Ero1. *Cell* 129 (2), 333–344
3. Reid GA, Miles CS, Moysey RK, Pankhurst KL, Chapman SK (2000). Catalysis in fumarate reductase. *Biochim Biophys Acta.* 1459 (2-3), 310-315.

6. There is mention of another paper for the crystallization/purification details. This in my opinion should be provided for a manuscript to a journal of this impact level.

Response: We agree with the reviewer's opinion. We included the purification data in this impact journal in the Methods section and in Supplementary Fig 1.

Reviewer: 3

We thank the reviewer for constructive comments on our work

Comments:

1. Although the authors do not address this directly, Supplementary Table 1 appears to indicate that this second FAD molecule or, more precisely the FMN portion of it that could be modelled is associated with much higher B-factors. In addition, this partial modelling of FMN would indicate that the affinity of FMN to Osm1 should be higher than that of FAD, which is contradicted by the findings in Supplementary Table 3.

Response: In our manuscript, we show the second FAD binding site on Osm1, which as previously mentioned is a novel and unexpected result. Our crystal structure showed only an FMN moiety at the second FAD binding site despite having added a high amount of FAD. Based on the structure, which shows the importance of an FMN moiety for FAD binding to Osm1, we hypothesized that other flavin molecules such as FMN and riboflavin could behave as alternative electron transfer molecules. From the enzyme reaction, as noted by the reviewer, Osm1 oxidized free FMN as well as FAD (Fig. 3E). Interestingly, the limited second FAD binding circumstance composed of several aromatic residues on Osm1, which could explain the lower affinity of the second FAD and function of Osm1, was used to make pi-pi stacking interactions with the only isoalloxazine ring of FAD. This interaction strategy explains that the second FAD binding site should serve as a relatively low-affinity site to allow for the efficient exchange with residual reduced flavin molecules, while the first FAD binding site could provide a high-affinity site to inhibit the exchange of FAD. We discuss this in the Discussion section. Although this study may not provide an explanation for the higher affinity of FAD compared to FMN at second FAD binding site of Osm1, what is shown is that flavin molecules possessing an isoalloxazine ring are able to bind to the second FAD binding site with low affinity.

2. The discussion mirrors the impression from the results section, the first half re-stating the obvious (flavin binding domain and clamp domain). If the conclusion is that the structure determined resembles the closed form, it is not understandable why the comparison in Supplementary Table 2 contains the less-closed form of *W. succinogenes* rather than the more-closed form available since 2001.

Response: The open form of Fcc3 (1QO8) had a higher Z-score even though the RMSD was 4.0. Based on this higher Z-score, the similarity was marked by the DALI server. The reason why the open form of Fcc3 or fumarate reductase sometimes has a higher Z-score is due to the partial domain structure, which includes a functionally important major part, as compared with the search model. We conducted another search to make sure that the open-form was included and used the best five matches. This new search also confirmed that the open-form has a high structural similarity with Osm1.

3. The second half of the discussion is predominantly composed of words like "... seems ... appears ... suggest ... can ... could ... could ... can ... may ... might ... could ... may ... may ... may ... may ..." and therefore hardly convincing.

Response: We have taken the reviewer's concern into consideration and have modified several sentences using more confirmative words. Our original text wanted to express the uncertainty of the novel mechanism found in our study.

REVIEWERS' COMMENTS:

Reviewer #1 (Remarks to the Author):

This reviewer believes that the paper has significantly been improved with additional data, figures and discussions and considerable rewriting of the text and therefore is now acceptable for publication in Nature Communications.

Reviewer #2 (Remarks to the Author):

This manuscript has been carefully and very positively reviewed to the various reviewers comments. New data and figures have been added, while some interpretations have been altered in response to the reviewers initial comments.

I have no further comments.